# Revaccination in Age-Risk Groups with Sputnik V Is Immunologically Effective and Depends on the Initial Neutralizing SARS-CoV-2 IgG Antibodies Level

**DOI:** 10.3390/vaccines11010090

**Published:** 2022-12-30

**Authors:** Mikhail A. Godkov, Darya A. Ogarkova, Vladimir A. Gushchin, Denis A. Kleymenov, Elena P. Mazunina, Evgeniia N. Bykonia, Andrei A. Pochtovyi, Valeriy V. Shustov, Dmitry V. Shcheblyakov, Andrey G. Komarov, Alexander N. Tsibin, Vladimir I. Zlobin, Denis Y. Logunov, Alexander L. Gintsburg

**Affiliations:** 1Department of Laboratory Diagnostics of N.V. Sklifosovsky Research Institute for Emergency Medicine, Moscow 129090, Russia; 2Department of Clinical Laboratory Diagnostics of FSBEI FPE “Russian Medical Academy of Continuous Professional Education” of the Ministry of Healthcare of the Russian Federation, Moscow 125993, Russia; 3Federal State Budget Institution “National Research Centre for Epidemiology and Microbiology Named after Honorary Academician N. F. Gamaleya” of the Ministry of Health of the Russian Federation, Moscow 123098, Russia; 4Department of Virology, Biological Faculty, Lomonosov Moscow State University, Moscow 119234, Russia; 5Moscow Healthcare Department, The Moscow City Government, Moscow 127006, Russia; 6Department of Infectiology and Virology, Federal State Autonomous Educational Institution of Higher Education I.M. Sechenov, First Moscow State Medical University of the Ministry of Health of the Russian Federation (Sechenov University), Moscow 119991, Russia

**Keywords:** COVID-19, vaccines, Sputnik V, Gam-COVID-Vac, Sputnik light, booster, immunological effectiveness, elderly people

## Abstract

Vaccination against COVID-19 has occurred in Russia for more than two years. According to the Russian official clinical guidelines to maintain tense immunity in the conditions of the ongoing COVID-19 pandemic, it is necessary to use booster immunization six months after primary vaccination or a previous COVID-19 contraction. It is especially important to ensure the maintenance of protective immunity in the elderly, who are at risk of severe courses of COVID-19. Meanwhile, the immunological effectiveness of the booster doses has not been sufficiently substantiated. To investigate the immunogenicity of Sputnik V within the recommended revaccination regimen and evaluate the effectiveness of booster doses, we conducted this study on 3983 samples obtained from individuals previously vaccinated with Sputnik V in Moscow. We analyzed the level of antibodies in BAU/mL three times: (i) six months after primary immunization immediately before the booster (RV), (ii) 3 weeks after the introduction of the first component of the booster (RV1), and (iii) 3 weeks after the introduction of the second component of the booster (RV2). Six months after the primary vaccination with Sputnik V, 95.5% of patients maintained a positive level of IgG antibodies to the receptor-binding domain (RBD) of SARS-CoV-2. The degree of increase in the specific virus-neutralizing antibodies level after revaccination increased with a decrease in their initial level just before the booster dose application. In the group of people with the level of antibodies up to 100 BAU/mL six months after the vaccination, a more than eightfold increase (*p* < 0.001, Wilcoxon criterion with Bonferroni adjustment) in the level of specific antibodies was observed (Me = 8.84 (IQR: 3.63–30.61)). A significant increase in the IgG level after receiving both the first and the second booster doses occurred at the initial titer level up to 300 BAU/ mL (*p* < 0.001) in those who did not contract COVID-19 in the past and up to 100 BAU/mL (*p* < 0.001) in those who were previously infected with SARS-CoV-2. A significant increase in the antibody level after the first dose of the booster was noted for people who had up to 500 BAU/mL (*p* < 0.05), regardless of the previous COVID-19 infection. Thus, revaccination is most effective in individuals with an antibody level below 500 BAU/mL, regardless of the vaccinee age and COVID-19 contraction. For the first time, it has been shown that a single booster dose of the Sputnik vaccine is sufficient to form a protective immunity in most vaccinees regardless of age and preexisting antibody level.

## 1. Introduction

The emergence of a new coronavirus at the end of 2019 in China led, within a few months, to the global spread of a previously unknown disease caused by the originally named 2019-nCoV virus [1]. Later, the new coronavirus was named SARS-CoV-2, and the disease it causes in humans is COVID-19 [2]. In terms of the scale of the COVID-19 pandemic, the SARS-CoV-2 virus has become the most significant infectious agent of the last century. In less than three years, more than 647 million cases of infection and 6.6 million deaths have been registered [3].

Vaccination against COVID-19 in Russia began on 11 August 2020 from the moment of the admission of the Sputnik V (Gam-COVID-Vac) vaccine into civil circulation according to the results of the 1/2 phase of clinical trials [4,5]. Sputnik V is a two-component adenovirus vector vaccine carrying 26 and 5 serotypes of adenovirus in the first and second component, respectively, encoding the SARS-CoV-2 S-antigen. Two adenoviruses are injected in three-week intervals, which allows for the achievement of a high level of viral neutralizing antibodies and cellular immunity, including CD4+ and CD8+ components [4]. At the first stage, vaccination was recommended to people from risk groups, including medical workers, transport workers, and trade workers who are in contact with a large number of people potentially infected with SARS-CoV-2. However, according to the results of the third phase studies, which demonstrated the high safety profile and epidemiological effectiveness of the Sputnik V [6], the vaccine was recommended for use in mass vaccination programs in Russia and 71 countries with a total population of 4 billion people [7]. According to the design of the third phase study, which included risk groups (over 60 years old) and to the results of studies among the elderly, the vaccine was shown to be highly effective, not significantly differing in immunogenicity and epidemiological efficacy from that of younger age groups. So, the vaccine was widely recommended for use among the elderly as well.

Mass vaccination with Sputnik V was launched in early 2021 among all residents over 18 years old except for those with some concomitant diseases. [8,9]. The peak of vaccination in Moscow occurred in the summer of 2021. However, in this period, it became obvious that to maintain tense immunity in relation to the SARS-CoV-2 Delta variant and other WHO-recognized VOCs (variants of concern), it was necessary to introduce booster doses six months after primary immunization [10]. Due to the results obtained for other vaccines using the SARS-CoV-2 S-antigen as the main immunogen, as well as the observation of a decrease in viral neutralizing activity against the Delta variant [11], the Russian ministry of health provisionally introduced a booster immunization six months after the initial immunization or COVID-19 contraction. Such recommendations have been approved in the Russian official clinical guidelines on vaccination to maintain tense immunity in the conditions of the ongoing COVID-19 pandemic [12].

In the summer of 2021, Sputnik Light became available in the Russian Federation for vaccination against COVID-19 [13]. It is the first component of the Sputnik V vaccine, where an adenovirus 26th serotype is used to deliver the S-antigen. The epidemiological effectiveness of the Sputnik Light vaccine during the spread of the Delta variant was studied using data from official registers of those who were vaccinated and infected with COVID-19 [14]. In groups of people under 60 years old, the epidemiological effectiveness was more than 78%, but in the group of older people, it was only 50%. A high effectiveness of Sputnik Light for revaccination was shown for people who had a COVID-19 infection on anamnesis [15]. That study involved only unvaccinated individuals and there were no age-risk groups. Despite the fact that a number of patients who were previously infected with SARS-CoV-2 were included in the study, it did not take into account the dependence of the effectiveness of booster doses on the initial level of specific virus neutralizing antibodies. In another study, it was shown that after vaccination with Sputnik V in medical workers, the level of antibodies significantly decreased 5–6 months after vaccination [16]. However, it should be noted that the study was limited in the sample for dynamic observation (n = 94 people), as well as the lack of published data on the dynamics of the level of virus-neutralizing antibodies in risk groups, considering the recommended vaccination and revaccination regimen with Sputnik V and Sputnik Light vaccines.

In this work, we investigated the immunological effectiveness of Sputnik V in various age groups as well as dependence on the level of virus-neutralizing antibodies at the time of booster application.

## 2. Materials and Methods

### 2.1. Study Design

Since the recommended revaccination regimen according to official Russian clinical guidelines implied the use of booster doses six months after primary immunization [12], the main objectives of our study were (i) to assess the effectiveness of preserving virus-neutralizing IgG antibodies to the receptor-binding domain (RBD) of the Spike protein of SARS-CoV-2 six months after the primary vaccination, (ii) to investigate the immunological effectiveness of the booster vaccination by Sputnik V depending on age, and (iii) to determine the immunological effectiveness of the booster vaccination by Sputnik V depending on the initial antibodies level. In the case of adenoviral vaccines, a hypothetical risk associated with revaccination is a possible antiadenovirus antibody reducing the booster effectiveness [17]. To answer these questions, a study was conducted with participants from 30 organizations in Moscow. All these institutions are designed to provide social services in stationary or semistationary forms.

Overall, 3983 patients aged 20 to 103 years old who were vaccinated and revaccinated with Sputnik V were included in this study. There were 1754 men (44%) with an average age of 64.19 ± 13.65 (95% CI: 63.55–64.83) years old and 2229 women (56%) with an average age of 73.07 ± 13.47 (95% CI: 72.51–73.63) years old. Thus, a significant proportion of the studied individuals were in a risk group according to age. Since the Sputnik V vaccine is a two-component vaccine, patients were vaccinated using both components according to the instructions [9]. The vaccines were applied intramuscularly in the forearm. The interval between the administration of the first and second components of the Sputnik V was 23.17 ± 2.37 (95% CI: 23.09–23.24) days. After six months from the first dose, the patients were revaccinated (the average time between the introduction of the first component during vaccination and the first component during revaccination was 190.23 ± 0.04 days (95% CI: 190.15–190.31 days). Since the approved revaccination scheme in half a year does not differ from the primary vaccination scheme, in order to study the immunological effectiveness of the revaccination, blood sera were taken three times: (i) before the introduction of the first revaccination component (RV1), as well as (ii) three weeks after the RV1 and (iii) three weeks after the second component of revaccination (RV2).

### 2.2. Collection of Serum Samples and Analysis

The diagnosis of COVID-19 was confirmed by the detection of SARS-CoV-2 RNA using a molecular diagnostic method based on a reverse transcription polymerase chain reaction in real time or with the use of computed tomography with the determination of the characteristics of the COVID-19 foci of infection in the lungs. Before vaccination and revaccination, all the patients underwent a medical examination with an explanation of possible side effects. The vaccine was received intramuscularly in the forearm. The interval between the introduction of the first and second components of Sputnik V was 21 days.

Written informed consent was obtained from all subjects in accordance with the order of the Ministry of Health of the Russian Federation of 21 July 2015 #474 n. This study was reviewed and deemed exempt by the Local Ethics Committee of the Gamaleya Center (protocol No. 14, 29 September 2021).

The test system for the quantitative analysis of the virus neutralizing antibodies Anti-SARS-CoV-2 IgG Quantitative (Ortho-Clinical Diagnostics, Illkirch-Graffenstaden, France) was used. The average time between the end of the vaccination and the first analysis was 159.99 ± 2.73 days (95% CI: 159.90–160.07). The level of IgG antibodies to RBD was measured in international units, i.e., BAU/mL (binding antibody units per milliliter), according to the WHO standard for the quantitative determination of neutralizing antibodies [18].

### 2.3. Description of Vaccination Groups

At the time of the first analysis, 752 people (18.88%) contracted COVID-19 in the past, 693 of whom (17.40%) had been ill once before vaccination, 10 (0.25%) had been ill once after vaccination. 46 people (1.23%) were ill twice, of which 38 (0.95%) were ill both times before vaccination and only 8 (0.20%) were ill the first time before vaccination and the second time after. A total of 3 (0.08%) people were ill three times (they were ill twice before vaccination and once after). All combinations of COVID-19 and vaccination anamnesis distributions are shown in Figure 1.

Three groups were created depending on the COVID-19 disease vaccination status distribution (Figure 1), which included (i) people who did not contract COVID-19 during the study, named Group 1; (ii) people who contracted COVID-19 before vaccination (regardless of the number of times), named Group 2; and (iii) people who contracted COVID-19 after vaccination (regardless of the presence or absence of COVID-19 in anamnesis before vaccination and regardless of the number of COVID-19 episodes), named Group 3 (Figure 1). The first group included 3231 people, the second 731, and the third 21.

### 2.4. Statistical Analysis

The programs SPSS Statistics, ver. 26 (IBM, Armonk, NY, USA); RStudio; and GraphPad Prism 8 (San Diego, CA, USA) were used for the statistical analysis. All quantitative parameters were examined for normality using the Kolmogorov–Smirnov test (n > 50) or Shapiro–Wilk test (n ≤ 50). For distributions that differed from normal (*p* < 0.05), the median and interquartile range were used to describe the findings (Me(IQR)), and nonparametric tests were used to compare groups with each other: Mann–Whitney, Kruskal-Wallis, Wilcoxon, and Friedman’s two-factor rank analysis of variance. The criterion used is indicated in specific tables. For parameters whose distribution did not differ from normal (*p* > 0.05), an average with a 95% confidence interval was used for description and parametric tests to compare groups (explanations given in the text).

When checking the multiplicity of the antibody level increase after each stage of revaccination, the increase level for each volunteer was calculated and subsequently the characteristics of the resulting distribution were described. The increase in the antibody level was determined as the antibodies ratio equal to the antibodies level after RV1 or RV2 divided by the initial antibodies level.

A regression analysis was used to more accurately determine the level of antibodies at which a multiple increase in the level of antibodies would occur. A linear relationship was assumed and verified between the natural logarithm of the enhancement coefficient and age, COVID-19 disease history, and the initial antibody level. A regression analysis was carried out using SPSS Statistics ver. 26. To characterize the quality of the obtained models, the Pearson correlation coefficient, the coefficient of determination R^2^, and the Fisher criterion were used.

## 3. Results

### 3.1. Antibodies Level Analysis Six Months after Vaccination

The antibodies level just before revaccination according to the groups described in the materials and methods section was studied. Group 1 included 3231 people who did not have COVID-19 before and during the observation period. Group 2 included 731 people who contracted COVID-19 only before vaccination, and Group 3 included 21 people who had COVID-19 after the primary vaccination. The results of the virus-neutralization antibody level analysis are presented in Table 1. The time elapsed between the first component during the primary vaccination and the analysis in Group 1 was slightly less than in the Group 2 (*p* < 0.001); however, the differences were less than one day and amounted to 190.34 ± 0.05 (95% CI: 190.25–190.43) days and 189.76 ± 0.07 (95% CI: 189.62–189.62) days, respectively (Table 1).

People from Group 2 were slightly older than in the Group 1 (*p* < 0.001). The median age in Group 1 was 69 (63–79) years, and in Group 2 it was 71 (64–82) years. The tendency to increase in age persisted in the Group 3 (the median was 75 (69–83) years). This may indicate a greater risk of the COVID-19 disease with age.

Six months after the primary vaccination with Sputnik V, 95.5% of the vaccinated individuals maintained a positive level of IgG antibodies to RBD (BAU/mL > 15, n = 3804). The level of antibodies in 21.8% of the patients was < 100 BAU/mL > (n = 870), 34.3% < 200 BAU/mL > (n = 1366), and 43.1% < 300 BAU/mL (n = 1715). The antibodies level in Group 1 was significantly lower than in Group 2 (*p* < 0.001). In Group 1, the median was 372 (116.5–1045) BAU/mL, while in Group 2 it was 553 (149.5–1950) BAU/mL (Figure 2a). The tendency to increase remained for Group 3, with the median level of antibodies being 747 (199–2250) BAU/mL; however, the statistical significance of the increase in this group was not found, probably due to the small size of this group (n = 21).

Six months after the primary vaccination, a tendency to increase the level of antibodies with age was noticed. The younger age groups (up to 60 years old) seemed to have lower levels of antibodies (Figure 2b). The antibodies level in Group 2 in all age groups was significantly higher (*p* < 0.05, Figure 2b) than in Group 1, except for the > 90 year old cohort. For this group, the level of antibodies was close, with a tendency to decrease in Group 2 (Figure 2b).

### 3.2. Immunological Effectiveness of Revaccination with Sputnik V Depending on Age

In order to study the effectiveness of revaccination, we used data on 3,959 people from Group 1 and Group 2 (according to Table 1) for analysis. People who contracted COVID-19 after the revaccination were excluded from the analysis (n = 3). The results demonstrated a significant increase in the antibodies level after revaccination in Group 1 in all age cohorts except for those >90 years old. In this cohort, the initial antibody level before revaccination was highest with the median 712.5 BAU/mL (239–1690) (n = 146) (Table 2). For almost every age cohort (≤90 years old), the increase in the number of antibodies was statistically significant both after the introduction of the first component and after the introduction of the second component. The exception was the age cohort of 71–80 years old, for which the RV2 did not lead to a statistically significant increase in the antibodies level (*p* = 0.677).

For Group 2, there was also a general increase in the antibodies level; however, it was less sharp than that among the people from Group 1. So, for all age cohorts (except cohorts 71–80 and >90 years), a significant increase in the antibodies level was characteristic after completion of the booster immunization. However, these differences were not significant when analyzing the increase in the antibodies level after receiving each component separately (Table 2). For the age cohorts 71–80 and >90 years old, it was not possible to detect a significant increase in the level of antibodies after the booster immunization; however, an increasing tendency was present in the 71–80 years old group. Thus, in almost all groups ≤90 years old regardless of COVID-19 history, revaccination increased the level of virus-neutralizing antibodies.

### 3.3. Immunological Effectiveness of Vaccination with Sputnik V Depending on the Initial Level of Antibodies before Revaccination

To study the effectiveness of revaccination in groups with different levels of antibodies before revaccination, all study participants were arbitrary divided into cohorts according to the level of antibodies detected before revaccination in the ranges 0–100, 100–200, 200–300, 300–400, 400–500, 500–2000, and 2000–4000 BAU/mL. For these cohorts, according to the level of antibodies before vaccination, an increase in the level of antibodies after the first and second injections with Sputnik V during revaccination was investigated. The study of the effectiveness of the revaccination was conducted separately for Group 1 and Group 2 (according to Table 1). To exclude the influence of test errors with results approaching the limits of sensitivity, only samples from patients with baseline levels below 4000 BAU/mL were examined during the analysis. In total, data on samples obtained from 3623 people were analyzed.

Among those in Group 1, there was a clear tendency for the antibody level before revaccination to increase with age from 66 in the 0–100 BAU/mL cohort to 73 in the 2000–4000 BAU/mL cohorts (*p* < 0.001, Kruskal–Wallis criterion) (Table 3). In the cohorts of Group 2, this tendency persisted, starting with an antibody level 100–200 BAU/mL (from 65 to 74 years, *p* < 0.001). In the 0–100 BAU/mL group, the median age was significantly higher than in the 100–200 BAU/mL group and was 73 years (*p* < 0.001). 

The most pronounced increase in antibody levels occurred in the cohort of people with antibody levels 0–100 BAU/mL before revaccination (Table 3, Figure 3). The increase in antibody levels after RV1 and RV2 was 5.04 and 8.84 times for those from Group 1 and 12.83 and 32.60 for those from Group 2. For cohorts of people who had antibody levels of 100–200 and 200–300 from Group 1, a significant increase occurred on both doses, while for cohorts of people who had 300–500 BAU/mL, the level of antibodies increased only on the first jab, although at a fairly low level not exceeding 1.5 times the increase. For cohorts with 100–500 BAU/mL from Group 2, elevated antibody levels on the first jab only were observed. In general, the fold increase was inversely proportional to the initial level of antibodies before revaccination (Table 3).

For people with an initial antibody level of 500–2000 BAU/mL from Group 2 (Table 3, Figure 3), no significant antibody level increase was observed (*p* = 0.730, Friedman’s test). For people from Group 1, a significant difference in antibody titer was achieved only after the introduction of both components of the vaccine (*p* = 0.038). At the same time, even in this case, in half of the patients, the increase did not exceed 1.04 times. For the initial titer in the range of 2000–4000 BAU/mL, there was a slight decrease in the level of antibodies after revaccination, both after the first and second jab (*p* < 0.001) for both the Group 1 and Group 2 people. The median of the increase factor in both cases was below one.

For finer estimations, the cohorts of 500–1000, 1000–1500, and 1500–2000 BAU/mL were analyzed. As a result, the boosting effect up to 1000 BAU/mL was maintained for the first jab of the Group 1 people (*p* = 0.033) (Appendix A), and no statistically significant increase was found for the Group 2 people.

For a more accurate assessment of the level of residual immunity at which revaccination may be recommended, a regression analysis was performed. For this purpose, patients with antibody levels <1000 BAU/mL were selected, and the seroconversion rate was evaluated after the first jab of the vaccine relative to the level of antibodies before revaccination in those who were from Group 1 (n = 2391) and Group 2 (n = 453). The age in Group 1 was in the range of 20–103 years old and the median was 68 (IQR: 61–76) years old, and in Group 2 it was 21–97 years old and the median was 69 (IQR: 63–81) years old (*p* = 0.003, Mann–Whitney criterion).

A linear regression model was used to evaluate the dependence of the natural logarithm of the increase coefficient on age, COVID-19 disease history, and the initial titer of the volunteer’s antibodies. The result is shown in Equation (1):(1)$$\ln\left(k_{1}\right)=0.781+0.011X_{age}+0.294X_{group}-0.003X_{AT}$$
where *k*_1_ is the antibodies ratio equal to the antibodies level after the first revaccination component divided by the number of antibodies before revaccination, *X_age_* is the age of the volunteer (years), *X_group_* is the history of COVID-19 (1—if were ill, 0—if were not ill), and *X_AT_* is the initial titer of the antibodies (BAU/mL).

This model was significant (*p* < 0.001) and took into account 24.9% of the factors determining the antibodies ratio, according to the determination coefficient. The model was characterized by a Pearson correlation coefficient of 0.499 (*p* < 0.001, a noticeable relationship according to the Chaddock scale).

According to the definition that was used earlier, a fourfold increase in the titer of antibodies at the time of analysis compared to the initial level of antibodies can be considered a successful seroconversion [6]. However, this is relevant if the comparison occurs with immune-naive people. We assumed that in the case of revaccination, a twofold increase in the level of antibodies after the first component of revaccination could be enough. So, we calculated the antibodies level at which we can recommend at least a single revaccination as:$$X_{AT}=\frac{0.781+0.011X_{age}+0.294X_{group}-\ln\left(2\right)}{0.003}$$

For a 65-year-old person without COVID-19 history, the threshold value was 268 BAU/mL, and for a person with COVID-19 history, the threshold value was 366 BAU/mL.

With a similar analysis of the antibody’s ratio after the second component of revaccination compared with the initial level, the following Equation (2) was obtained:(2)$$\ln\left(k_{2}\right)=1.268+0.011X_{age}+0.253X_{group}-0.003X_{AT}$$
where *k*_2_ is the antibodies ratio equal to the antibodies titer after the second component of revaccination divided by the antibody titer before the start of revaccination.

Equation (2) is also significant (*p* < 0.001), took into account 30.6% of the factors determining the antibodies ratio, and was characterized by a Pearson correlation coefficient r = 0.554 (*p* < 0.001, a noticeable relationship according to the Chaddock scale).

For 65-year-old people from Group 1, the threshold value was 430 BAU/mL, and for people from Group 2, the threshold value was 514 BAU/mL, at which we could expect a twofold increase in antibodies after the introduction of the second component. A fourfold increase in antibodies after the introduction of the second component can be expected, with an initial level not higher than 199 BAU/mL and 283 BAU/mL for people from Group 1 and Group 2, respectively. Figure 4 shows the dependence of the threshold level of antibodies on age, antibody titer before vaccination, and COVID-19 contraction in the anamnesis. Since the age of most of the volunteers who participated in the study was in the range from 61 to 83 years, these ages are shown in Figure 4.

The results obtained generally correlate with the results in Table 3 and confirm that there was a decrease in the effectiveness of revaccination with an increase in the initial titer of antibodies before revaccination; however, the regression gave a more positive assessment for effective revaccination and higher thresholds of the initial titer of antibodies for recommending booster immunization.

## 4. Discussion

Studies show that booster doses of COVID-19 vaccines increase the virus-neutralizing activity of antibodies, enhances the effectiveness of vaccines in protecting against the symptomatic COVID-19, and prevents hospitalization. The positive effects of boosters are shown for the Delta [10,19] and Omicron SARS-CoV-2 variants, which has been the main variant of concern over the past year [20]. Six different booster vaccines from AstraZeneca, Curevac, Johnson & Johnson (Janssen, Beerse, Belgium), Moderna, Novavax, Pfizer, and Valneva boosted antibody and neutralizing responses after the adenoviral ChAd/ChAd (from AstraZeneca) initial course and most with no safety concerns. [10]. It seems that it is especially important to use boosters to increase antibody capacity against Omicron. A near-complete lack of neutralizing activity against Omicron in polyclonal sera from individuals vaccinated with two doses of the BNT162b2 COVID-19 vaccine and from convalescent individuals was reported [20]. However, mRNA booster immunizations in vaccinated and convalescent individuals resulted in a significant increase in serum neutralizing activity against Omicron. This demonstrates that booster immunizations can critically improve the humoral immune response against the Omicron variant. Our previous results showed that the use of Sputnik V as a booster to previous Sputnik V vaccination increases the protection from hospitalization against Omicron [21]. It was shown that revaccination with third and fourth doses of Sputnik V increased the effectiveness of protection. To make a decision on revaccination with Sputnik Light, a single booster immunization can be recommended if the dose is below a quantitative value of 142.7 BAU/mL, as was previously proposed [15]. In the frame of our study, this result was significantly extended along age cohorts and depending on the virus-neutralizing level before revaccination. It has been shown that the immunological effectiveness of revaccination increases with a decrease in the initial level of antibodies. We have constructed linear regression models indicating that the effectiveness of revaccination significantly depends on the initial titer and the age of the revaccinated. To achieve classical seroconversion with a fourfold increase in the level of neutralizing antibodies during revaccination after six months from primary vaccination, a 65-year-old person should have an initial level not higher than 199 BAU/mL if they have not had COVID-19 and 283 BAU/mL in case of COVID-19 contraction in their anamnesis. If we consider the sufficiency of a twofold increase in the level of antibodies, then the threshold value is 430 BAU/mL if they have not had COVID-19, and 514 BAU/mL in the case of COVID-19 contraction in their anamnesis. In general, cohorts of those who have been vaccinated with reduced antibody levels over time benefit most from the booster, regardless of the age and previous COVID-19 disease. The data obtained by us indicate that at least a one-time repeated use of both doses of Sputnik V is an effective means of increasing the level of virus-neutralizing antibodies for people under 90 years of age with an antibody titer below 500 BAU/mL.

The data obtained by us showed that after six months from the moment of full vaccination, 95.5% of patients maintained a positive test level for IgG antibodies to RBD (BAU/mL > 15). For comparison, in clinical studies, IgG seroconversion to RBD occurred in 98.25% [6]. In our study, the presence of higher antibody titers with age six months after vaccination was significant. In the third phase clinical study, such differences were not noticed. Unfortunately, the design of our study did not allow us to assess what level of antibodies was achieved immediately after the primary vaccination for the cohort we studied, but it allowed us to conclude that after 6 months, a large proportion of vaccinated people retain positive IgG antibodies to RBD (BAU/mL > 15).

It was previously shown that for 80% protection against infection with the Alpha variant (B.1.1.7), it is necessary to have an antibody level corresponding to 171 (95% CI: 57–519) BAU/mL [22]. Our study showed that the median antibody level after half a year was 372 (IQR 116.5–1045), 553 (149.5–1950), and 747 (199–2250) in Group 1 (without COVID-19 contraction), Group 2 (people with COVID-19 before primary vaccination), and Group 3 (people with COVID-19 after primary vaccination), respectively. In the total sample, the level of antibodies in 21.8% was < 100 (n = 870), 34.3% < 200 (n = 1366), and 43.1% < 300 BAU/mL (n = 1715). This may indicate that at least half of the individuals remain protected six months after the primary vaccination, while the other half may require booster jabs to maintain tense immunity.

Those who were previously infected with SARS-CoV-2 in all age cohorts had higher levels of antibodies and responded more actively to the administration of booster doses of Sputnik V. This may be due to the different clonotypes of B cells formed during revaccination and SARS-CoV-2 infection [23], which may lead to the formation of more effective hybrid immunity against COVID-19 [24]. Age cohorts showed that individuals younger than 60 years had significantly lower levels of antibodies, regardless of previous COVID-19 contraction. At the same time, both groups of vaccinees from Group 2 and Group 3 (according to the Table 1 division) had a slightly older age. One of the factors of the observed phenomenon may be the aging of the immune system of elderly vaccinees [25,26].

Revaccination efforts should be primarily aimed at patients whose antibody levels have decreased the most over time, as well as at-risk groups. Our study shows that revaccination is most effective in groups with the lowest level of antibodies six months after the primary vaccination. Additionally, the data obtained allowed us to propose a revaccination scheme with Sputnik V depending on the initial level of antibodies. Thus, revaccination at a response level above 500 BAU/mL does not give a significant increase in antibodies, and in this group, revaccination can be postponed to a later period. For patients with an antibody level 200–500 BAU /mL, revaccination gives an increase in the level of antibodies on average less than three times, and it will probably be advisable to use revaccination in this case in risk groups and with only one dose. For patients with an antibody level 0–200 BAU/mL, revaccination with two doses of Sputnik V can be used.

## 5. Study Limitations

Our study is not without limitations. This work was performed using one test system that allowed us to determine the level of IgG antibodies to the RBD domain of the Wuhan variant. However, this system has been validated for quantitative measurements of antibodies in international virus-neutralizing units of BAU/mL [18]. In addition, the vaccines used still use the S-protein variant of the original virus variant. Meanwhile, most countries still use the original versions of vaccines for revaccination. Since the threshold values of immunological markers of protection for new variants of the virus, primarily the Omicron variant, are not known, we cannot say how much revaccination allows us to achieve the necessary levels. Additionally, we have not investigated how effective the use of booster doses will be maintained with respect to subsequent revaccinations, implying fourth and fifth doses of Sputnik V. According to the current guidelines, booster immunizations are recommended in six months without a limit on antibody level. Additional studies of the effectiveness of subsequent doses are required. Additionally, we did not study the role of cellular immunity, which is certainly important for protection against COVID-19. Revaccination according to the open data registry in the Russian Federation has passed only 10% of the population, which indicates that the majority of people will require revaccination in the future. The patterns we discovered on the immunological efficacy of Sputnik V and Sputnik Light should remain relevant for most people in Russia.

## 6. Conclusions

The extent to which antibody levels increase after revaccination depends on the initial antibody level just before revaccination. Groups with the lowest levels of antibodies most actively responded to revaccination, but booster jabs are still effective with an initial antibodies level up to 500 BAU/mL. Although older vaccinees generally had higher antibody levels in our study cohort six months after the primary immunization, booster doses were effective across all age groups. A single booster dose of the Sputnik Light (first component of Sputnik V) vaccine is sufficient to form a tense immunity in most vaccinees regardless of age and preexisting antibody level. These factors can be taken into account when planning a revaccination company.

## Figures and Tables

**Figure 1 vaccines-11-00090-f001:**
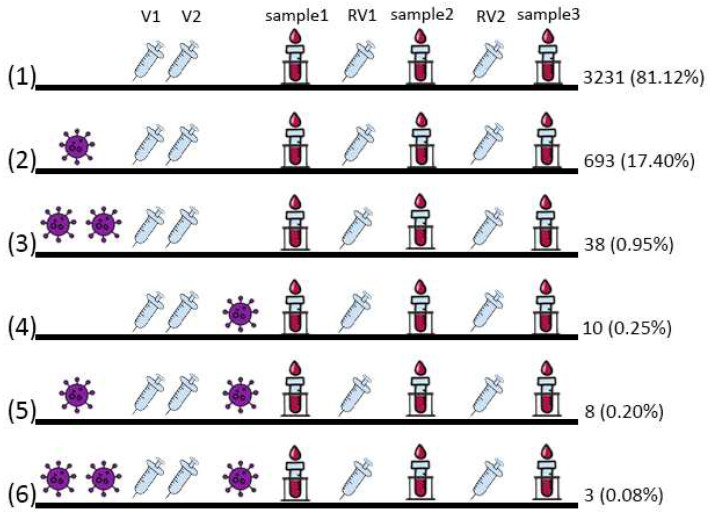
General distribution of vaccinations according to possible variants of the combination of COVID-19 contraction and vaccination in the first analysis: (1) did not get COVID-19; (2) got sick once before vaccination; (3) got sick twice before vaccination; (4) got sick once after vaccination; (5) got sick twice: first time before vaccination, the second time after; and (6) got sick three times: twice before vaccination and once after. The figure conventionally indicates the time of the disease, the time of vaccination (V1 and V2) and revaccination (RV1 and RV2), and the time of taking the analysis (sample 1, sample 2, and sample 3). The number of people and the proportion of the entire sample under study are given.

**Figure 2 vaccines-11-00090-f002:**
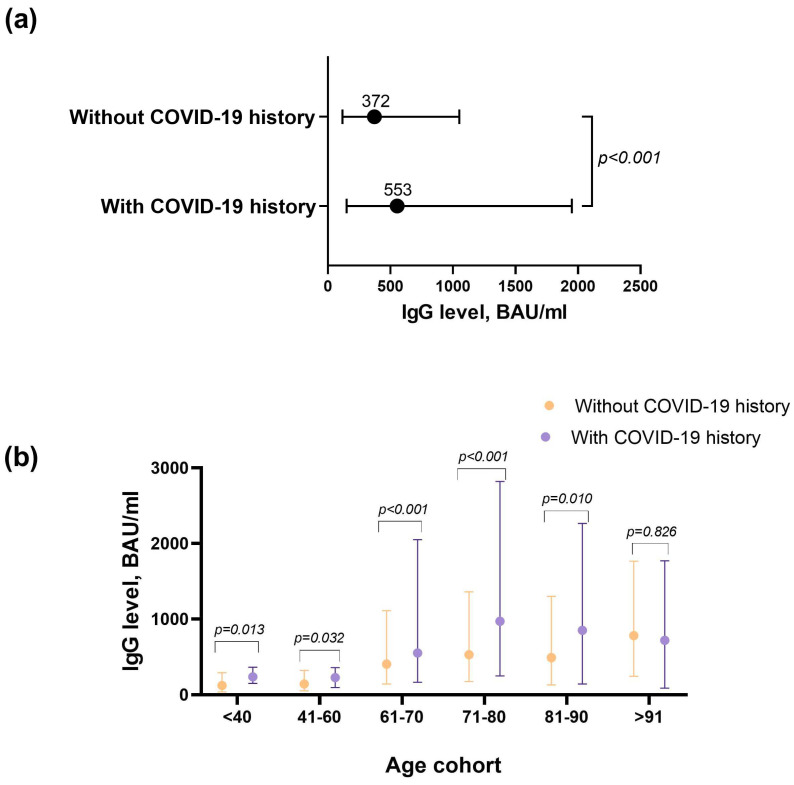
The level of antibodies six months after vaccination with Sputnik V, depending on age and the presence of COVID-19 in the anamnesis. (**a**) Median and interquartile range are shown without age division, (**b**) median and interquartile range are shown in groups by age and history of COVID-19. Only Groups 1 and 2 are included in the figure.

**Figure 3 vaccines-11-00090-f003:**
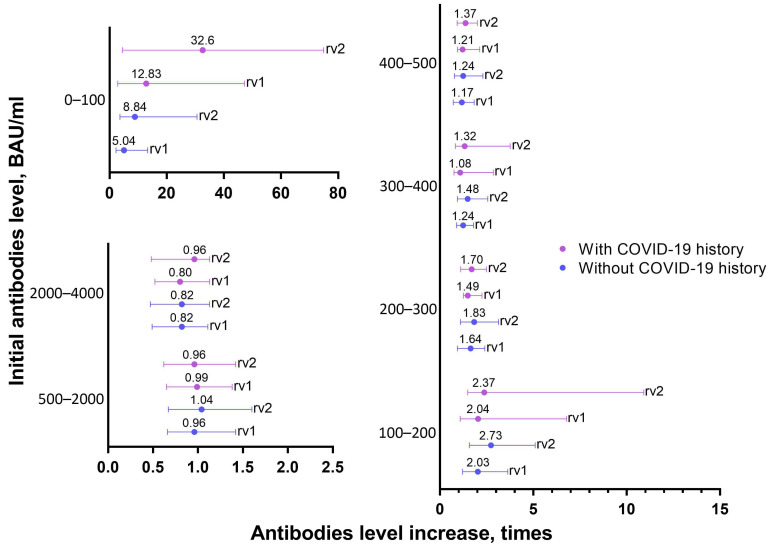
Increasing IgG to RBD level after booster vaccination, depending on the COVID-19 contraction in the anamnesis. In the figure, the marker shows the median of the antibodies ratio (the ratio of the antibody titer after the introduction of the first or second component of revaccination to the antibody titer before revaccination), and the whiskers show the interquartile range. The values above the graph show the level of antibodies before revaccination.

**Figure 4 vaccines-11-00090-f004:**
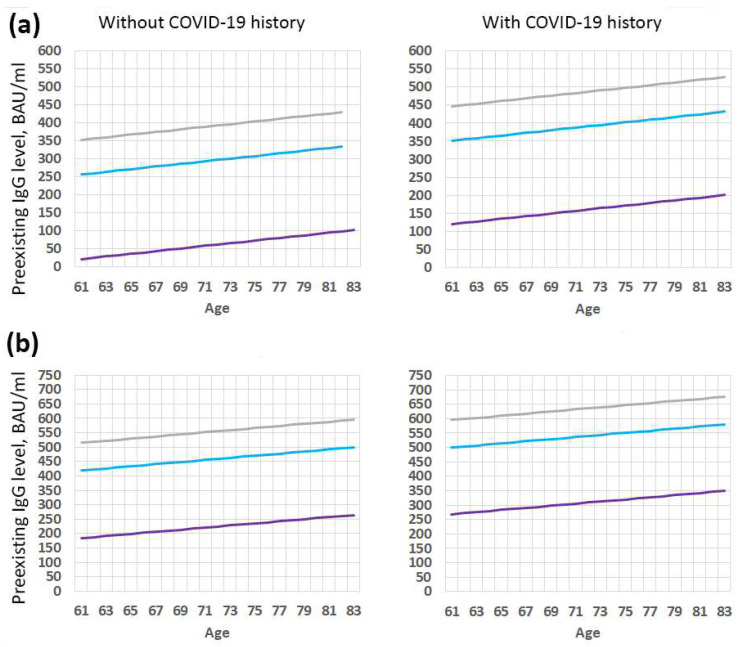
Threshold level of antibodies before the start of revaccination, at which (and below) an antibodies ratio will be predicted at least 1.5 (gray), 2 (blue), or 4 (purple) times after the introduction of the first (RV1) (**a**) or second (RV2) (**b**) jab during revaccination.

**Table 1 vaccines-11-00090-t001:** Demographic description and the level of IgG to RBD six months after immunization.

KERRYPNX	Group 1 without COVID-19 History n = 3231	Group 2 COVID-19 before Vaccination n = 731	Group 3 COVID-19 after Vaccination n = 21	*p*
The time between the Sputnik V vaccination and sampling RV for analysis M (95% CI)	190.34 ± 0.05 (190.25–190.43)	189.76 ± 0.07 (189.62–189.90)	189.52 ± 0.32 (188.85–190.19)	0.029 * (Student-s F-criterion) A posteriori comparison (Games–Howell criterion) *p*_12_ < 0.001 * *p*_13_ = 0.05 *p*_23_ = 0.761
Age, years Me (IQR)	69 (63–79)	71 (64–82)	75 (69–83)	<0.001 * (Kruskal–Wallis criterion) A posteriori comparison (Bonferroni adjustment) *p*_12_ < 0.001 * *p*_13_ = 0.143 *p*_23_ = 0.734
IgG, BAU/mL Me (IQR)	372 (116.5–1045)	553 (149.5–1950)	747 (199–2250)	<0.001 * (Kruskal–Wallis criterion) A posteriori comparison (Bonferroni adjustment) *p*_12_ < 0.001 * *p*_13_ = 0.300 *p*_23_ = 1.000
<40 ages old	n = 180	n = 33	--	
Me (IQR)	124.5 (38.9–294)	239 (150–367)		0.013 * (Mann–Whitney criterion)
41–60	n = 356	n = 58		
Me (IQR)	142.5 (53.6–324.5)	229.5 (97.3–362)		0.032 * (Mann–Whitney criterion)
61–70	n = 1203	n = 265		
Me (IQR)	405 (144–1110)	551 (166–2050)		0.001 * (Mann–Whitney criterion)
71–80	n = 818	n = 162		
Me (IQR)	528 (175–1360)	970 (252–2820)		0.001 * (Mann–Whitney criterion)
81–90	n = 527	n = 168		
Me (IQR)	490 (131–1300)	850.5 (143.5–2265)		0.010 * (Mann–Whitney criterion)
>90	n = 147	n = 45		
Me (IQR)	780 (247–1765)	719 (88.6–1770)		0.826 (Mann–Whitney criterion)
*p*	<0.001	<0.001		
	*p*_<40vs61–70_ < 0.001 *	*p*_<40vs61–70_ = 0.013 *		
	*p*<_40vs71–80_ < 0.001 *	*p*_<40vs71–80_ = 0.001 *		
	*p*<_40vs81–90_ < 0.001 *	*p*_<40vs81–90_ = 0.005 *		
	*p*_<40vs>90_ < 0.001 *			
	*p*_41–60vs61–70_ < 0.001 *	*p*_41–60vs61–70_ < 0.001 *		
	*p*_41–60vs71–80_ < 0.001 *	*p*_41–60vs71–80_ < 0.001 *		
	*p*_41–60vs81–90_ < 0.001 *	*p*_41–60vs81–90_ < 0.001 *		
	*p*_41–60vs>90_ < 0.001 *	*p*_41–60vs>90_ = 0.031 *		
	*p*_61–70vs>90_ = 0.009 *			

* The differences are significant (*p* < 0.05). p_12_ shows significance of the differences for Group 1 and Group 2, *p*_13_ shows significance of the differences for Group 1 and Group 3, *p*_23_ shows significance of the differences for Group 2 and Group 3.

**Table 2 vaccines-11-00090-t002:** Immunological effectiveness of revaccination with Sputnik V six months after primary vaccination with Sputnik V, depending on COVID-19 anamnesis and age.

	Antibody Level before Revaccination (RV), Me (IQR)	Antibody Level after RV1, Me (IQR)	Antibody Level after RV2, Me (IQR)	*p* (Friedman’s Analysis). The Pairwise Comparison Are Corrected by Bonferroni Adjustment
Without COVID-19 history (Group 1)				
<40, n = 180	124.5 (38.9–294)	268.5 (168–379)	332.5 (193–488.5)	<0.001 * *p*_12_ < 0.001 * *p*_13_ < 0.001 * *p*_23_ = 0.025 *
41–60, n = 356	142.5 (53.6–324.5)	237.5 (135–356.5)	349 (222–568)	<0.001 * *p*_12_ < 0.001 * *p*_13_ < 0.001 * *p*_23_ < 0.001 *
61–70, n=1203	405 (144–1110)	651 (296–1300)	761 (355.5–1585)	<0.001 * *p*_12_ < 0.001 * *p*_13_ < 0.001 * *p*_23_ = 0.001 *
71–80, n = 818	528 (175–1360)	726 (359–1480)	806.5 (400–1870)	<0.001 * *p*_12_ < 0.001 * *p*_13_ < 0.001 * *p*_23_ = 0.677
81–90, n = 526	487.5 (131–1300)	641.5 (272–1570)	842 (400–1800)	<0.001 * *p*_12_ < 0.001 * *p*_13_ < 0.001 * *p*_23_ = 0.007 *
>90, n = 146	712.5 (239–1690)	712.5 (225–2000)	826.5 (280–1720)	0.722
With COVID-19 history (Group 2)				
<40, n = 33	239 (150–367)	307 (193–386)	404 (186–526)	0.035 * *p*_12_ = 0.804 *p*_13_ = 0.029 * *p*_23_ = 0.419
41–60, n = 58	229.5 (97.3–362)	344.5 (235–414)	413.5 (220–1050)	0.013 * *p*_12_ = 0.061 *p*_13_ = 0.021 * *p*_23_ = 1.000
61–70, n = 265	551 (166–2050)	986 (395–2570)	964 (428–2940)	<0.001 * *p*_12_ = 0.007 * *p*_13_ < 0.001 * *p*_23_ = 0.214
71–80, n = 162	970 (251–2820)	1075 (397–3220)	1370 (579–3930)	0.150
81–90, n = 167	857 (143.5–2265)	1310 (673–2740)	1430 (738.5–3465)	<0.001 * *p*_12_ = 0.002 * *p*_13_ < 0.001 * *p*_23_ = 1.000
>90, n = 45	719 (88.6–1770)	1140 (564–1880)	911 (584–1850)	0.232

* The differences are significant (*p* < 0.05). p_12_ shows statistical significance of the differences between RV and RV1, p_13_ shows statistical significance of the differences between RV and RV2, p_23_ shows statistical significance of the differences between RV and RV2.

**Table 3 vaccines-11-00090-t003:** Immunological effectiveness of revaccination with Sputnik V depending on initial antibodies level before revaccination.

	Antibody Level before Revaccination (BAU/mL)	Age, Years Me (IQR)	Antibody Level BAU/mL		*p* (Wilcoxon Criterion with Bonferroni Adjustment)	Antibody Level Increasing, Times (Antibody Ratio)	
			After RV1, Me (IQR)	After RV2, Me (IQR)		After RV1, Me (IQR)	After RV2 Me (IQR)
Without COVID-19 history (Group 1)							
Low initial level of IgG to RBD	0–100, n = 721	66 (55–76)	207 (86–411)	325 (153–810)	*p*_12_ < 0.001 * *p*_13_ < 0.001 * *p*_23_ < 0.001 *	5.04 (2.24–13.33)	8.84 (3.63–30.61)
	100–200, n = 427	66 (58–74)	294 (175.5–518)	386 (226.5–714)	*p*_12_ < 0.001 * *p*_13_ < 0.001 * *p*_23_ < 0.001 *	2.03 (1.21–3.63)	2.73 (1.57–5.10)
Medium initial level of IgG to RBD	200–300, n = 295	67 (62–75)	394 (236–583)	435 (275.5–777)	*p*_12_ < 0.001 * *p*_13_ < 0.001 * *p*_23_ = 0.008 *	1.64 (0.93–2.401)	1.83 (1.10–3.14)
	300–400, n = 228	68 (62–75)	441 (316–614)	515 (328.5 – 877)	*p*_12_ < 0.001 * *p*_13_ < 0.001 * *p*_23_ = 1.000	1.24 (0.89–1.80)	1.48 (0.93–2.56)
	400–500, n = 180	69 (63–76.5)	508.5 (315–813)	555.5 (341.5–967.5)	*p*_12_= 0.032 * *p*_13_ = 0.003 * *p*_23_ = 1.000	1.17 (0.71–1.83)	1.24 (0.77–2.30)
High initial level of IgG to RBD	500–2000, n = 962	72 (65–80)	919.5 (542–1450)	964 (603–1640)	*p*_12_ = 0.547 *p*_13_ = 0.038 * *p*_23_ = 0.735	0.96 (0.66–1.42)	1.04 (0.67–1.60)
Very high initial level of IgG to RBD	2000–4000, n = 198	73 (67–82)	2085 (1280–3390)	2175 (1370–3430)	*p*_12_< 0.001 * *p*_13_ = 0.001 * *p*_23_ = 1.000	0.82 (0.49–1.11)	0.82 (0.47–1.13)
With COVID-19 history before vaccination (Group 2)							
Low initial level of IgG to RBD	0–100, n = 145	73 (66–82)	385 (137–1180)	953 (206–2110)	*p*_12_ < 0.001 * *p*_13_ < 0.001 * *p*_23_ < 0.001 *	12.83 (2.86–47.18)	32.60 (4.56–74.92)
	100–200, n = 67	65 (61–72)	291 (153–1014)	393 (204.5–1545)	*p*_12_ < 0.001 * *p*_13_ < 0.001 * *p*_23_ = 0.426	2.04 (1.08–6.78)	2.37 (1.49–10.92)
Medium initial level of IgG to RBD	200–300, n = 53	65 (50–72)	390 (311–573)	469 (257–611)	*p*_12_ < 0.001 * *p*_13_ < 0.001 * *p*_23_ = 1.000	1.49 (1.26–2.25)	1.70 (1.11–2.49)
	300–400, n = 52	67 (60–75)	379 (260.5 – 950.5)	445 (296–1260)	*p*_12_ = 1.000 *p*_13_ = 0.001 * *p*_23_ = 0.026 *	1.08 (0.75–2.86)	1.32 (0.82–3.76)
	400–500, n = 34	70 (63.5–81.5)	553.5 (406–958)	607 (428–829)	*p*_12_ = 0.011 * *p*_13_ = 0.011 * *p*_23_ = 1.000	1.21 (0.93–2.13)	1.37 (0.92–2.00)
High initial level of IgG to RBD	500–2000, n = 201	74 (66–83)	1020 (678–1410)	993 (619–1590)	*p*_12_ = 1.000 *p*_13_ = 1.000 *p*_23_ = 1.000	0.99 (0.65–1.38)	0.96 (0.62–1.42)
Very high initial level of IgG to RBD	2000–4000, n = 68	74 (64.5–83)	2175 (1445–3305)	2175 (1290–3495)	*p*_12_ = 0.014 * *p*_13_ = 0.030 * *p*_23_ =1.000	0.80 (0.52–1.13)	0.96 (0.48–1.13)

* The differences are significant (*p* < 0.05). p_12_ shows statistical significance of the differences between RV and RV1, p_13_ shows statistical significance of the differences between RV and RV2, p_23_ shows statistical significance of the differences between RV1 and RV2.

## Data Availability

Inquiries about access to the original clinical data should be directed to the corresponding author.

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
