# Peer review of "Revaccination in Age-Risk Groups with Sputnik V Is Immunologically Effective and Depends on the Initial Neutralizing SARS-CoV-2 IgG Antibodies Level"

_vaccines, 2022, doi:10.3390/vaccines11010090_

Round 1

Reviewer 1 Report

Dear authors, interesting idea but the study is written in a confusing manner that after reading it for at least three times I am still not clear on many things. Below are some suggestions . 

Abstract:

What does RBD stands for in line 42 ?

Introduction:

Line 94 ?  previously undergone COVID-19 ? I think author ment to say they are previously infected with COVID-19 . Please correct if this is the case.

Methods: line 115 , might want to add 3,983 instead of 3 983 , please correct it in the manuscript else . where. Use similar formatting for all the numbers every where in the manuscript.

Results :  Authors has present results in very confusing manner and need major revision and write in a format it is understandable

Section 3.1

Please describe the full form of RBD line 157 or above in line 42

Section 3.1 from lin e155 to 164 should be included in methods section and not in results section.

Also paragraph 165 to 179 have many components of study design that should be in methods section and not here.

Please rewrite the methods section properly describing study design. only results should be in the results section.

Section 3.2

Line 173 -174 – average time was six months but in ur study it was 167.07 with CI of 2.66 ? can you explain .

Line 177- 180 – I am not abel to understand when actually blood samples were collected . It need to be clearly explained here.

1-1   ( 3 weeks apart ) primary vaccine nation series ? – so one sample was collected at before 3 weeks ?

Second sample was collected 3 weeks after the shot 2 nd ?  It is not clear and very confusing

Again, the description of groups will go in study design not here. Line 196 to 200 .

Line 197; - So in group 1 Authors mention person who were not ill before the first analysis. ( Does this group include people who got ill after the first analysis ? – then how this group I is different from group II ? _ please explain or clarify as  )

Line 200; formatting - 3 231 people, the second - 731, the third – 21 ? remove dashes before number , does it means minus 731 ? minus 21 ?  

Line 203 – line 209 . Authors need to format numbers every where , “,” is used at many  places where “ . “ should be used , please fix it everywhere. Ideally this should have been done before submission. Same for the table which is also not numbered.

In table below n what is that authos have included in brackets ? for example what does 124,5 [38,9 – 294] stands for ?

Table 1 the antibody level is I think before the third dose ? I am unsure

Line 213 – 216 ? < 100 , < 200 and < 300 numbers are mutually exclusive ? if yes then right rage here

Line 216 and 218 – the level of antibodies in group who were vaccinated and were ill with COVID-19 turned out to be significantly lower than in the group of those who had been ill before vaccination?  What does first statement states I am not abel to understand .

Figure 2 b , can u use contrasting colour , I can not make difference in two groups !!!

Section 3.3

Where is table 1? Label it

Line 236 to 238 - Why to combine these two groups, the data should be studied separately in these tow groups. Did authors combine these two groups? Sems like they haven’t , please clarify the statement here.

 Table 2 fix the “,” with “.” – should have been done before submission

3.3 line 264 to 266

why authors choose these groups ? its arbitrary or there is some scientific evidence behind it that group with < 100 cmpared to > 500 have worse or poor outcomes ? if yes cite the study here or describe the reason .

Discussion :

379 – 382 – not sure what they meant by this statement. can the author please clarify?

The discussion seems rushed and the authors did not discuss all the findings in the results clearly. 

Author Response

Thank you for valuable comments concerning the manuscript. We provided the additional information and made appropriate corrections along the text.

Response to Reviewer 1 Comments

Dear authors, interesting idea but the study is written in a confusing manner that after reading it for at least three times I am still not clear on many things. Below are some suggestions .

Thank you for finding our research interesting. We have improved it according to your suggestions and recommendations!

Abstract:

What does RBD stands for in line 42 ?

RBD means Receptor Binding Domain of SARS-CoV-2 virus, the part of Spike protein. Thank you for your suggestion, we corrected it now.

Introduction:

Line 94 ?  previously undergone COVID-19 ? I think author ment to say they are previously infected with COVID-19 . Please correct if this is the case.

Thanks for the comment, we corrected it in the text now. Indeed we mean a group of people who contracted COVID-19 in the past.

Methods: line 115 , might want to add 3,983 instead of 3 983 , please correct it in the manuscript else . where. Use similar formatting for all the numbers every where in the manuscript.

We have corrected the format of numbers for a more comfortable perception. We replaced spaces with commas.

Results :  Authors has present results in very confusing manner and need major revision and write in a format it is understandable

Thank you for your suggestion! We have redesigned the results structure to make it easier to understand.

Section 3.1

Please describe the full form of RBD line 157 or above in line 42

Corrected now!

Section 3.1 from lin e155 to 164 should be included in methods section and not in results section.

Also paragraph 165 to 179 have many components of study design that should be in methods section and not here.

Please rewrite the methods section properly describing study design. only results should be in the results section.

Thank you for your comments! We have changed materials and methods and omitted the  information from the results section.

Section 3.2

Line 173 -174 – average time was six months but in ur study it was 167.07 with CI of 2.66 ? can you explain .

Thank you for your comment! The design of the study assumed revaccination six months after the introduction of the first component, the paper indicates the time after the second component, which is approximately 21-23 days less. We agree that this causes misunderstanding and corrected the terms of revaccination from the first component in the article. So, the average time between the introduction of the first component during vaccination and the first component during revaccination was 190.23±0.04 days (95% CI: 190.15 – 190.31 days).

Line 177- 180 – I am not abel to understand when actually blood samples were collected . It need to be clearly explained here.

It has been corrected now to simplify the perception!

1-1   ( 3 weeks apart ) primary vaccine nation series ? – so one sample was collected at before 3 weeks ?

Second sample was collected 3 weeks after the shot 2 nd ?  It is not clear and very confusing

The first blood sample was collected before revaccination by the first component of Sputnik V. The second blood sample was collected three weeks after receiving the first component (before receiving the second component of Sputnik V) and the third analysis was collected three weeks after receiving  the second component. We have slightly changed this sentence to make it clearer in the text, also we rewritten this part in material and methods.

Again, the description of groups will go in study design not here. Line 196 to 200

Thank you! We moved it to the materials and methods section now.

Line 197; - So in group 1 Authors mention person who were not ill before the first analysis. ( Does this group include people who got ill after the first analysis ? – then how this group I is different from group II ? _ please explain or clarify as  )

We explained groups more accurately using references to Figure 1. We improved Figure 1 and made it easy to understand. Thank you for your questions. It was really difficult to perceive.

Line 200; formatting - 3 231 people, the second - 731, the third – 21 ? remove dashes before number , does it means minus 731 ? minus 21 ? 

We corrected it now. Thank you.

Line 203 – line 209 . Authors need to format numbers every where , “,” is used at many  places where “ . “ should be used , please fix it everywhere. Ideally this should have been done before submission. Same for the table which is also not numbered.

We have corrected throughout

In table below n what is that authos have included in brackets ? for example what does 124,5 [38,9 – 294] stands for ?

This format stands for Median and interquartile range, we explained it in the first column or in the first row in each table. We have corrected throughout

Table 1 the antibody level is I think before the third dose ? I am unsure

We corrected misunderstandings. And added correct title to this table

Line 213 – 216 ? < 100 , < 200 and < 300 numbers are mutually exclusive ? if yes then right rage here

These values are not mutually exclusive, but are cumulative, that is, the number of individuals with an antibody level <200 includes those with an antibody level <100. Similar with <300 BAU/ml.

Line 216 and 218 – the level of antibodies in group who were vaccinated and were ill with COVID-19 turned out to be significantly lower than in the group of those who had been ill before vaccination?  What does first statement states I am not abel to understand .

We corrected that. Unfortunately, the preposition ‘not’ was lost during translation. In the group of those who were not ill (the first group), the median was 372 [116.5 – 1045] BAU/ml, and in those who were ill before vaccination (the second group) – 553 [149.5 – 1950] BAU/ml

Figure 2 b , can u use contrasting colour , I can not make difference in two groups !!!

Thank you! Indeed we corrected colors on this Figure 2b.

Section 3.3

Where is table 1? Label it

We labeled it now

Line 236 to 238 - Why to combine these two groups, the data should be studied separately in these tow groups. Did authors combine these two groups? Sems like they haven’t , please clarify the statement here.

In this section, we compared groups 1 and 2 and excluded group 3 from the comparison, since there were very few people in it. But we did not combine groups 1 and 2 and studied it separately.

 Table 2 fix the “,” with “.” – should have been done before submission

 We corrected that. Thank you. We will be more careful next time.

3.3 line 264 to 266

why authors choose these groups ? its arbitrary or there is some scientific evidence behind it that group with < 100 cmpared to > 500 have worse or poor outcomes ? if yes cite the study here or describe the reason .

Indeed, there are studies and even reviews that show that the protective level of antibodies is in the range of 100-200. We cite one of the papers ref [18] in the discussion [Perry, J.; Osman, S.; Wright, J.; Richard-Greenblatt, M.; Buchan, S.A.; Sadarangani, M.; Bolotin, S. Does a Humoral Correlate of Protection Exist for SARS-CoV-2? A Systematic Review. PLoS One 2022, 17, 1–20, doi:10.1371/journal.pone.0266852.]. In this regard, the division according to the level of antibodies was mechanistic at the first stage to determine which of the groups according to the level would respond with a statistically significant increase in antibodies. However, we then carry out additional analysis using regression analysis. There, the values at which the level rises by a given value are already determined without using predetermined artificially selected groups.

Discussion :

379 – 382 – not sure what they meant by this statement. can the author please clarify?

We changed this phrase to simplify perception on “Studies show that the use of booster doses of COVID-19 vaccines increases the virus-neutralizing activity of antibodies, enhances the effectiveness of vaccines in protecting against the symptomatic course of COVID-19 and prevents hospitalization. …

The discussion seems rushed and the authors did not discuss all the findings in the results clearly.

Thank you! We have improved the discussion section now. Your comments were very useful! We tried to improve English language too but we will use MDPI English service for language polishing after paper acceptance too.

Reviewer 2 Report

1) Abstract. We analyzed the level of antibodies in BAU /ml three times: (i) six months after primary immunization immediately before booster (RV), (ii) 3 weeks after the introduction of the first component of the booster (RV1) (iii) 3 weeks after the introduction of the second component of the booster (RV2). Six months after vaccination with Sputnik V, 95.5% of patients maintained a positive level of IgG antibodies to RBD. The multiplicity of the increase in the level of specific neutralizing antibodies after revaccination increased with a decrease in the initial level of antibodies before revaccination. In the group with the level of antibodies up to 100 BAU/ml six month after the vaccination, a more than fivefold increase in the titer of specific antibodies was observed after revaccination. A significant increase in the titer after the receiving both the first and the second booster doses occurs at the initial titer level up to 300 BAU / ml in those who were not previously ill and up to 200 BAU /ml in those who had previously undergone COVID-19. A significant increase of the antibody level after the first dose of booster is noted for people who had up to 1000 BAU / ml, regardless of the history of COVID-19. Could you please add the most important statistical values to support the data.
2) Abstract. Revaccination is most effective in individuals with an antibody level below 1000 BAU/ml, regardless of the COVID-19 history and the patients’ age. A single booster dose of the vaccine is sufficient to form a tense immunity in most patients. Please underline the novelty of the study.
3) 1. Introduction L56-58. Vaccination against COVID-19 in Russia began on August 11, 2020 from the moment  of admission of the Sputnik V (Gam-Covid-Vac) vaccine into civil circulation according to  the results of the 1/2 phase of clinical trials [1,2]. Could you please add a brief introduction on COVID 19. Furthermore, in order to discuss these described points, important references are needed to be added, such as:
a) Genomic Biomarker Heterogeneities between SARS-CoV-2 and COVID-19. Vaccines 202210, 1657. https://doi.org/10.3390/vaccines10101657
b) Interstitial Lung Disease at High Resolution CT after SARS-CoV-2-Related Acute Respiratory Distress Syndrome According to Pulmonary Segmental Anatomy. J Clin Med. 2021;10(17):3985. Published 2021 Sep 2. doi:10.3390/jcm10173985
c) Effects of Prior Infection with SARS-CoV-2 on B Cell Receptor Repertoire Response during Vaccination. Vaccines 202210, 1477. https://doi.org/10.3390/vaccines10091477
d) Different Methods to Improve the Monitoring of Noninvasive Respiratory Support of Patients with Severe Pneumonia/ARDS Due to COVID-19: An Update. J Clin Med. 2022;11(6):1704. Published 2022 Mar 19. doi:10.3390/jcm110617044. 
4) Introduction. L106-109. For this purpose, we examined blood serum samples obtained from 3 983 individuals, including many patients from age-risk groups.  Further, the level of antibodies in these patients was analyzed after vaccination by two  components of Sputnik V, depending on age, as well as the initial level of antibodies. Please, these sentences are part of method.
5) 2. Materials and Methods 2.1 Collection of Serum Samples and Analysis L112-114. The study included patients from 30 medical organizations in Moscow. All of them  are designed to provide social services in stationary, semi-stationary forms of social services on the terms of permanent, temporary or five-day round-the-clock residence of citizens. During the study period, 3 983 people were under observation. Please add some information on the study populations.
6) Discussion L378-381. Studies show that in the case of booster doses of vaccines, the neutralizing ability of antibodies, the effectiveness of protection against symptomatic COVID-19, as well as preventing the need for hospitalization, which is typical as in the case of the Delta variant  [6,15], and in the case of Omicron variants [16] increases. Please, summarise here the most important results of the study
7) Discussion. L 441-446. Our study is not without limitations. The work was performed using one test system  that allows determining the level of IgG antibodies to the RBD domain of the Wuhan  variant. However, this system has been validated for quantitative measurement of  antibodies in international virus neutralizing units BAU/ml. In addition, the vaccines used  still use the S-protein variant of the original virus variant. Meanwhile most countries still  use the original versions of vaccines for revaccination. Please add a separate paragraph on study limits.
8) Discussion. Please, add a brief paragraph of conclusions.

Author Response

Thank you for valuable comments concerning the manuscript. We provided the additional information and made appropriate corrections along the text.

Response to Reviewer 2 Comments

1) Abstract. We analyzed the level of antibodies in BAU /ml three times: (i) six months after primary immunization immediately before booster (RV), (ii) 3 weeks after the introduction of the first component of the booster (RV1) (iii) 3 weeks after the introduction of the second component of the booster (RV2). Six months after vaccination with Sputnik V, 95.5% of patients maintained a positive level of IgG antibodies to RBD. The multiplicity of the increase in the level of specific neutralizing antibodies after revaccination increased with a decrease in the initial level of antibodies before revaccination. In the group with the level of antibodies up to 100 BAU/ml six month after the vaccination, a more than fivefold increase in the titer of specific antibodies was observed after revaccination. A significant increase in the titer after the receiving both the first and the second booster doses occurs at the initial titer level up to 300 BAU / ml in those who were not previously ill and up to 200 BAU /ml in those who had previously undergone COVID-19. A significant increase of the antibody level after the first dose of booster is noted for people who had up to 1000 BAU / ml, regardless of the history of COVID-19. Could you please add the most important statistical values to support the data.

Thanks for the recommendation. We added the p-value and data on the statistical criteria used to demonstrate the statistical significance of the results.

2) Abstract. Revaccination is most effective in individuals with an antibody level below 1000 BAU/ml, regardless of the COVID-19 history and the patients’ age. A single booster dose of the vaccine is sufficient to form a tense immunity in most patients. Please underline the novelty of the study.

We have rewritten the sentence to underline the novelty as “For the first time it has been shown that a single booster dose of the Sputnik V vaccine is sufficient to form a tense immunity regardless age and preexisting antibody level.”

3) 1. Introduction L56-58. Vaccination against COVID-19 in Russia began on August 11, 2020 from the moment  of admission of the Sputnik V (Gam-Covid-Vac) vaccine into civil circulation according to  the results of the 1/2 phase of clinical trials [1,2]. Could you please add a brief introduction on COVID 19. Furthermore, in order to discuss these described points, important references are needed to be added, such as:

  1. a) Genomic Biomarker Heterogeneities between SARS-CoV-2 and COVID-19. Vaccines 2022, 10, 1657. https://doi.org/10.3390/vaccines10101657
  2. b) Interstitial Lung Disease at High Resolution CT after SARS-CoV-2-Related Acute Respiratory Distress Syndrome According to Pulmonary Segmental Anatomy. J Clin Med. 2021;10(17):3985. Published 2021 Sep 2. doi:10.3390/jcm10173985
  3. c) Effects of Prior Infection with SARS-CoV-2 on B Cell Receptor Repertoire Response during Vaccination. Vaccines 2022, 10, 1477. https://doi.org/10.3390/vaccines10091477
  4. d) Different Methods to Improve the Monitoring of Noninvasive Respiratory Support of Patients with Severe Pneumonia/ARDS Due to COVID-19: An Update. J Clin Med. 2022;11(6):1704. Published 2022 Mar 19. doi:10.3390/jcm110617044.

We have improved introduction brief description on COVID 19 and started with: “The emergence of a new coronavirus at the end of 2019 in China led within a few months to the global spread of a previously unknown disease caused by the originally named 2019-nCoV virus [Li Q. et al. Early Transmission Dynamics in Wuhan, China, of Novel Coronavirus-Infected Pneumonia. // N Engl J Med. 2020. Vol. 382, № 13. P. 1199–1207.]. Later, the new coronavirus was named SARS-CoV-2, and the disease it causes in humans is COVID-19 [Naming the coronavirus disease (COVID-19) and the virus that causes it [Electronic resource]. URL: https://www.who.int/emergencies/diseases/novel-coronavirus-2019/technical-guidance/naming-the-coronavirus-disease-(covid-2019)-and-the-virus-that-causes-it (accessed: 05.12.2022).]. In terms of the scale of the COVID-19 pandemic, the SARS-CoV-2 virus has become the most significant infectious agent of the last century. In less than three years, more than 623 million cases of infection and 6.5 million deaths have been registered.”

One of the suggested references has been used in the discussion “c) Effects of Prior Infection with SARS-CoV-2 on B Cell Receptor Repertoire Response during Vaccination. Vaccines 2022, 10, 1477. https://doi.org/10.3390/vaccines10091477”

4) Introduction. L106-109. For this purpose, we examined blood serum samples obtained from 3 983 individuals, including many patients from age-risk groups.  Further, the level of antibodies in these patients was analyzed after vaccination by two  components of Sputnik V, depending on age, as well as the initial level of antibodies. Please, these sentences are part of method.

Thank you for your comment, we moved this sentence in the materials and methods section.

5) 2. Materials and Methods 2.1 Collection of Serum Samples and Analysis L112-114. The study included patients from 30 medical organizations in Moscow. All of them  are designed to provide social services in stationary, semi-stationary forms of social services on the terms of permanent, temporary or five-day round-the-clock residence of citizens. During the study period, 3 983 people were under observation. Please add some information on the study populations.

We added a new paragraph in the Materials and Methods section where we described some demographic characteristics of study populations and group forming.

6) Discussion L378-381. Studies show that in the case of booster doses of vaccines, the neutralizing ability of antibodies, the effectiveness of protection against symptomatic COVID-19, as well as preventing the need for hospitalization, which is typical as in the case of the Delta variant  [6,15], and in the case of Omicron variants [16] increases. Please, summarise here the most important results of the study summarise here the most important results of the study

We have rewritten this paragraph and summarized here the most important results of the studies

7) Discussion. L 441-446. Our study is not without limitations. The work was performed using one test system  that allows determining the level of IgG antibodies to the RBD domain of the Wuhan  variant. However, this system has been validated for quantitative measurement of  antibodies in international virus neutralizing units BAU/ml. In addition, the vaccines used  still use the S-protein variant of the original virus variant. Meanwhile most countries still  use the original versions of vaccines for revaccination. Please add a separate paragraph on study limits.

Thank you for the suggestion that we have added a separate study limitation paragraph.

8) Discussion. Please, add a brief paragraph of conclusions.

Thank you for the suggestion that we have added a separate paragraph with conclusion remarks.

Reviewer 3 Report

The authors investigated the immunological effectiveness after revaccination status in age-risk groups with Sputnik V vaccin. Yet, they evaluated the effect compared with previous SARS-Cov-2 titers in blood serum samples following different post-vaccination periods or after the COVID disease.Their study is well designed and based on an important   number     of 3 983 people . They , evaluated the level of antibodies  after vaccination based on two components of Sputnik V, depending on age, and the initial level of antibodies.They found that people who had previously a COVID in all age groups showed higher levels of antibodies and responded more actively to the administration of booster doses of the Sputnik V vaccin. Yet, younger people up to 60  years showed significantly lower levels of antibodies Also,a significant increase in the titer after receiving both the first and the second booster doses occurs. Revaccination showed most effective in people with  antibodies level below 1000 50 BAU/ml, regardless of the COVID-19 history and the patients’ age.The paper is well written and the material and methods section well designed and approved by Ethical Committees. Results are evaluated following statistical analysis followed on an extended discussionbased on un updated bibliography .

My suggestion is to ACCEPT and publish the paper in its present form

Author Response

Thank you for valuable comments concerning the manuscript. We provided the additional information and made appropriate corrections along the text.

Response to Reviewer 3 Comments

The authors investigated the immunological effectiveness after revaccination status in age-risk groups with Sputnik V vaccin. Yet, they evaluated the effect compared with previous SARS-Cov-2 titers in blood serum samples following different post-vaccination periods or after the COVID disease.Their study is well designed and based on an important   number     of 3 983 people . They , evaluated the level of antibodies  after vaccination based on two components of Sputnik V, depending on age, and the initial level of antibodies.They found that people who had previously a COVID in all age groups showed higher levels of antibodies and responded more actively to the administration of booster doses of the Sputnik V vaccin. Yet, younger people up to 60  years showed significantly lower levels of antibodies Also,a significant increase in the titer after receiving both the first and the second booster doses occurs. Revaccination showed most effective in people with  antibodies level below 1000 50 BAU/ml, regardless of the COVID-19 history and the patients’ age.The paper is well written and the material and methods section well designed and approved by Ethical Committees. Results are evaluated following statistical analysis followed on an extended discussionbased on un updated bibliography .

My suggestion is to ACCEPT and publish the paper in its present form

Thank you very much for your revision and suggestion on accepting our manuscript.

Round 2

Reviewer 1 Report

Dear Authors, thanks for incorporating my suggestion. The revised manuscript makes more sense for the readers and